# European List of Essential Medicines for Medical Education: a protocol for a modified Delphi study

Erik Donker [1,2] David Brinkman,[1,2] Milan Richir,[1,2] Paraskevi Papaioannidou,[3] Robert Likic,[4,5] Emilio J Sanz [6] Thierry Christiaens,[7] João Costa,[8] Fabrizio De Ponti,[9] Milo Gatti,[9] Ylva Böttiger,[10] Cornelis Kramers,[11] Sarah Garner,[12] Rahul Pandit,[13] Michiel van Agtmael,[1,2] Jelle Tichelaar[1,2]

**Correspondence to**
Erik Donker;
e.donker@amsterdamumc.nl

## ABSTRACT

**Introduction** Junior doctors are responsible for a substantial number of prescribing errors, and final-year medical students lack sufficient prescribing knowledge and skills just before they graduate. Various national and international projects have been initiated to reform the teaching of clinical pharmacology and therapeutics (CP&T) during undergraduate medical training. However, there is as yet no list of commonly prescribed and available medicines that European doctors should be able to independently prescribe safely and effectively without direct supervision. Such a list could form the basis for a European Prescribing Exam and would harmonise European CP&T education. Therefore, the aim of this study is to reach consensus on a list of widely prescribed medicines, available in most European countries, that European junior doctors should be able to independently prescribe safely and effectively without direct supervision: the European List of Essential Medicines for Medical Education.

**Methods and analysis** This modified Delphi study will recruit European CP&T teachers (expert group). Two Delphi rounds will be carried out to enable a list to be drawn up of medicines that are available in ≥80% of European countries, which are considered standard prescribing practice, and which junior doctors should be able to prescribe safely and effectively without supervision.

**Ethics and dissemination** The study has been approved by the Medical Ethics Review Committee of VU University Medical Center (no. 2020.335) and by the Ethical Review Board of the Netherlands Association for Medical Education (approved project no. NVMO-ERB 2020.4.8). The European List of Essential Medicines for Medical Education will be presented at national and international conferences and will be submitted to international peer-reviewed journals. It will also be used to develop and implement the European Prescribing Exam.

## Strengths and limitations of this study

► To our knowledge, this will be the first study to reach consensus on a European List of Essential Medicines for Medical Education.
► The Delphi method is the most suitable method to reach consensus anonymously with a large group of experts working in different countries.
► The European List of Essential Medicines for Medical Education will help to harmonise clinical pharmacology and therapeutics (CP&T) education in Europe.
► There is already strong collaboration between European Association for Clinical Pharmacology and Therapeutics and WHO Europe, and the members are recognised experts in the field of CP&T education.
► Recruiting a sufficient number of participants from all European countries will be a challenge.

prescribing has become an increasingly complex task. International studies have shown that junior doctors are responsible for a substantial number of prescribing errors[1 3] and that at the time of graduation, junior doctors not only feel insufficiently prepared to prescribe safely and effectively but also have insufficient knowledge and skills to perform this task.[4–8]

This has prompted various national and international projects to reform teaching in clinical pharmacology and therapeutics (CP&T) in the undergraduate medical curriculum. For example, in the UK and the Netherlands a prescribing assessment has been introduced for final-year students, to ensure that they have acquired the necessary knowledge before graduation.[9–11] Other European countries might benefit from a similar initiative. As many countries do not have the time and resources to implement an assessment at a national level, we initiated an Erasmus+

## INTRODUCTION

Prescribing is a core task of junior doctors, who are responsible for most hospital prescriptions.[1 2] With an ever-expanding therapeutic arsenal and an increasing number of patients with comorbidity and polypharmacy, safe and effective

project (2019-1 - NL01 - KA203-060492) to develop, test and implement a standardised prescribing assessment on safe prescribing (including knowledge and skills) for undergraduate medical students studying at medical schools in the European Union (ie, European Prescribing Exam (EuroPE[+])). See http://www.prescribingeducation.eu/ for more information.

The assessment will be based on the 'essential diseases in prescribing' derived from a Delphi consensus study held in 2018.[12] However, there is no consensus list of medicines that European junior doctors should be able to independently prescribe safely and effectively without direct supervision. This list will form the basis of the European Prescribing Exam (especially the skills part), and together with country-specific adjustments to reform educational programmes in CP&T in all European countries. This will complement the wish of the European Association for Clinical Pharmacology and Therapeutics (EACPT) to harmonise European training in CP&T[13] and will be included in a future revision of the Guide to Good Prescribing of the WHO.[14] Previous studies of such lists have been based on the opinions of individuals or small groups of experts or were specific to one country.[15–18] Therefore, the aim of this study is to reach consensus on a list of medicines that are widely prescribed and available in Europe and which European junior doctors should be able to independently prescribe safely and effectively without direct supervision, that is, the European List of Essential Medicines for Medical Education.

## METHODS AND ANALYSIS

A modified Delphi method will be used as it has been shown to be an effective and successful method for reaching consensus on content of a CP&T curriculum.[12 19–23] As the availability of medicines differs between European countries, it is not possible to develop an all-encompassing list, but one that can be considered as a basis for European CP&T education. Subsequently, each country can adjust the list based on the availability of medicines in its country. The study will start in October 2020 and will comprise three phases: phase I—creating a drug list, selecting an European expert panel and developing a web-based questionnaire in Castor Electronic Data Capture; phase II—sending a questionnaire to appointed coordinators and phase III—Delphi consensus.

### Patient and public involvement
No patient involved.

### Phase I
#### Drug list
On the basis of the WHO Model List of Essential Medicines,[24] guideline therapies for the 'essential diseases in prescribing' (see online supplemental appendix

1), a literature review[15–18] and a drug list from the EuroPE[+] project,[25] a questionnaire will be developed regarding an extensive list of possible medicines that European junior doctors should be able to prescribe safely and effectively without supervision directly after graduation (online supplemental appendix 2). This list will be categorised into diseases. For each drug, the most commonly used routes of administration will be listed separately.

### Expert panel
Through the Education Working Group of the EACPT and the affiliated Network of Teachers in Pharmacotherapy, all coordinators (n=393) from all European medical schools (n=297) who are responsible for teaching CP&T to medical students will be approached to participate in the study. The coordinators will be asked to participate in the study themselves and to select a group of experts within their own centre, using the following criteria:

► Two experienced (≥3 years of teaching experience) teachers explicitly engaged in CP&T education for medical students, of which at least one teacher is registered as a clinical pharmacologist.
► At least five healthcare professionals, preferably a surgeon (eg, general surgeon); an internist (eg, general internist, gastroenterologists, pulmonologist and cardiologist); a general practitioner; a specialist in geriatric medicine or geriatrician and a (hospital) pharmacist.
► Two recently graduated junior doctors (graduated ≤1 year ago) who prescribe drugs on a daily basis.

There will be no restrictions regarding the work environment of the respondents (academic or community hospitals). The principal investigator will invite the experts to participate via email, providing an information letter and a link to the online survey. Prior to participation, the experts will be asked to sign a digital informed consent form.

Based on previous studies with this dedicated group of experts,[22 26] we expect a response rate of 25% for the coordinators, representing all European countries. Assuming 5–6 recruited experts per coordinator and a response rate of 25% as well, we expect in total 200–250 experts to complete the study.

### Phases II and III
#### Study design and data collection
In phase II, the coordinator(s) of each university will receive the list of medicines developed in phase I and will be asked to indicate which medicines are available in his or her country. On the basis of this information, a second questionnaire will be drawn up consisting of the medicines that are available in Europe. This questionnaire will be used for phase III, a two-round Delphi study. In round 1, all experts will be asked to evaluate two statements for each medicine (item): (1) 'In my country, it is standard practice to

prescribe this medicine to patients' and (2) 'A junior doctor should be able to independently prescribe this medicine safely and effectively without direct supervision'. Respondents will score both statements using a 5-point Likert scale (1=strongly disagree, 2=disagree, 3=neither agree nor disagree, 4=agree and 5=strongly agree). Respondents will also be able to add missing medicines that they consider should be included in round 2, and to add arguments for their choices in an open text box.

The questionnaire for round 2 will have the same structure as in round 1, but will also include the average group score per medicine from round 1, the suggested medicines and the arguments for their inclusion. The coordinator will be asked an additional question about the availability of the suggested medicines in his or her country: 'In my country, this medicine is available to prescribe to patients'.

To minimise participant drop-out, the list of medicines will be structured and participants are allowed to complete a portion of the survey and return later to finish the remaining part. The list must be completed within 2 weeks, a reminder will be send after 1 week.

## Statistics

A medicine will be included in the second questionnaire if that medicine is available in ≥80% of the European countries. In accordance with previous studies,[12 22] a medicine will be included in the final European List of Essential Medicines for Medical Education if both statements about this medicine are scored 4 or 5 by ≥80% of the respondents. If one statement about a medicine is scored 4 or 5 by ≥80% of the respondents but the other statement is scored 4 or 5 by ≥50%–<80% of the respondents, then this medicine will be reassessed in Delphi round 2. This also applies to suggested medicines. Medicines from round 2 will be included in the European List of Essential Medicines for Medical Education if both statements regarding a medicine are given a score of 4 or 5 by ≥80% of the respondents. Medicines suggested by respondents should also be available in ≥80% of European countries.

## ETHICS AND DISSEMINATION

Prior to participation, all experts will be asked to give their informed consent and provide the following information: email address, medical school, profession with background, and years of clinical and teaching experience. The data will be coded and stored for a maximum of 10 years in a secure folder on the hard disk of the Amsterdam UMC, location VUmc. Participation will not be professionally advantageous or disadvantageous and there will be no compensation for participation. Respondents can end their participation at any time, without giving a reason. The study has been approved by the Medical Ethics Review Committee of Amsterdam UMC, location VUmc (no. 2020.335) and by the Ethical Review Board of the Netherlands Association for Medical Education (approved project no. NVMO-ERB 2020.4.8).

The results of the study will be presented at national and international conferences and will be submitted to international peer-reviewed journals. The final European List of Essential Medicines for Medical Education will be used to develop and implement the European Prescribing Exam.

**Author affiliations**
[1]Department of Internal Medicine, Section Pharmacotherapy, Amsterdam UMC Locatie VUmc, Amsterdam, The Netherlands
[2]Research and Expertise Centre in Pharmacotherapy Education (RECIPE), Amsterdam, The Netherlands
[3]Department of Pharmacology, School of Medicine, Faculty of Health Sciences, Aristotle University of Thessaloniki, Thessaloniki, Greece
[4]University of Zagreb School of Medicine, Zagreb, Croatia
[5]Department of Internal Medicine, Unit of Clinical Pharmacology, University Hospital Centre Zagreb, Zagreb, Croatia
[6]School of Health Science, Universidad de La Laguna, La Laguna, Tenerife, Spain
[7]Department of Clinical Pharmacology, Ghent University, Gent, Belgium
[8]Department of Pharmacology and Clinical Pharmacology, University of Lisbon, Lisbon, Portugal
[9]Department of Medical and Surgical Sciences, Pharmacology Unit, Alma Mater Studiorum, University of Bologna, Bologna, Italy
[10]Department of Medical and Health Sciences, Linköping University, Linköping, Sweden
[11]Department of Internal Medicine and Pharmacology-Toxicology, Radboud University Medical Center, Nijmegen, The Netherlands
[12]Health Technologies and Pharmaceuticals Programme, WHO Regional Office for Europe, Copenhagen, Denmark
[13]Department of Translational Neuroscience, University Medical Centre Utrecht Brain Centre, Utrecht, The Netherlands

**Contributors** ED, DB, MR and JT: involved in devising the study design and writing the protocol. TC, YB, FDP, RL, CK, JC, EJS, PP, MvA, MG, RP and SG: reviewed and approved the manuscript.

**Funding** This study was funded by Erasmus+, grant number 2019-1 - NL01 - KA203-060492.

**Competing interests** SG reports her employment at WHO Europe.

**Patient consent for publication** Not required.

**Provenance and peer review** Not commissioned; externally peer reviewed.

**ORCID iDs**
Erik Donker http://orcid.org/0000-0002-8169-0714
Emilio J Sanz http://orcid.org/0000-0001-6788-4435

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
