## [Reviewer comments · BMJ Open]

ARTICLE DETAILS

TITLE (PROVISIONAL)	The European List of Essential Medicines for Medical Education, a Protocol for a Modified Delphi Study
AUTHORS	Donker, Erik; Brinkman, David; Richir, Milan; Papaioannidou, Paraskevi; Likic, Robert; Sanz, Emilio; Christiaens, Thierry; Costa, Joao; De Ponti, Fabrizio; Gatti, Milo; Böttiger, Ylva; Kramers, Cornelis; Garners, Sarah; Pandit, Rahul; van Agtmael, Michiel; Tichelaar, Jelle

VERSION 1 – REVIEW

REVIEWER	Garraud, Olivier University of Lyon
REVIEW RETURNED	06-Dec-2020

GENERAL COMMENTS	This study is a proposal to create a list of essential medicines that can be safely prescribed by junior doctors without supervision. It is based on a consensus panel and plans to us a
--

REVIEWER	Kinston, Ruth Keele University, School of Medicine
REVIEW RETURNED	07-Dec-2020

GENERAL COMMENTS	Thank you for asking me to review this protocol for a modified Delphi study with the research aim of developing a consensus list of medications that junior doctors should be able to prescribe safely and effectively without direct supervision. It was also made clear how such a list could form the basis of required knowledge/content of a European PSA and harmonise CP&T training. Methodology - I was reassured to see the scope and ambition of the intended recruitment of participants to this study (297 medical schools, 393 co-ordinators who in turn would recruit 9 expert participants). I was also reassured that the intended expert group would include representation from a broad range of clinical specialties and also include newly qualified doctors in their numbers. I was perplexed as to why pharmacists were not included (given the multi-professional nature of the prescribing process and their role in surveillance and safety) especially as they have been successfully included in studies where consensus was sort on conditions that junior doctors should be able to prescribe for. I wonder if this had been considered? I was also unclear about the aimed recruitment targets for the expert response and wondered if this could be clarified?
--

	Does 200-250 experts relate to the response rate from co-ordinators and their expert groups or does it relate to the total number of individual survey respondents (including specialists, junior doctors etc). Obviously this is likely to make a significant impact on the representative nature and consensus achieved. I wonder if this might be clarified? The protocol for each phase of the Delphi process and consensus thresholds used seemed clear to me and I felt the described process could be reasonably replicated. The limitations identified seemed reasonable but the papers applicability might to a large part depend on the initial recruitment (ensuring it was representative of the broadest range of European countries) and avoiding participant drop out between surveys (which may be related to elements like survey length etc). I would have welcomed more in the methods as to the implications and how this would be addressed. Overall I think the proposed study tries to seek consensus on a really important subject and I think the method chosen is entirely suitable if it can be executed to a high standard.
--	---

REVIEWER	Newby, David The University of Newcastle, Pharmacy and Experimental Pharmacology
REVIEW RETURNED	18-Dec-2020

GENERAL COMMENTS	Well written protocol. I just have a couple of typo/queries/suggestions: Page 9: The Drug List could be described before the Expert Panel simply to report this in the same order as referred to in Line 9 of page 30 (or change the order here) Page 16 Line 3 - unclear why there is a closing bracket at the end of 'Omeprazole, pantoprazole,esomeprazole' Page 18 Line 22 - unclear why it is called 'Folium acid' when in the previous list it was appropriately called 'Folic acid' Page 19 Line 26 - I think Zink should be Zinc Page 19 Line 27 - unclear why a brand (Vaseline) is used instead of the generic (petroleum jelly) Page 19 - miconazole appears twice under cutaneous (Line 22 and line 39)
--

VERSION 1 – AUTHOR RESPONSE

Reviewer 1

First of all thank for reviewing our protocol. Unfortunately, we did not receive your personal comments. The editor has send us the following, and we tried to response on it.

1. The reviewer has indicated that more clarity is needed around the framing of the research question, and more detail is needed in the methodology to enable the study to be replicated.

In the current manuscript we described the aim as follow: *“The aim of this study is to reach consensus on a list of medicines that are widely prescribed and available in Europe and which European junior doctors should be able to independently prescribe safely and effectively without direct supervision, i.e. The European List of Essential Medicines for Medical Education.”* We can frame this aim, and thus our research question, by stating that it should be medicines that **all** junior doctors should be able to prescribe **right after graduating from medical school**. We have adjusted the aim as follow:

“The aim of this study is to reach consensus on a list of medicines that are widely prescribed and available in Europe and which all European junior doctors right after graduation from medical school should be able to independently prescribe safely and effectively without direct supervision, i.e. The European List of Essential Medicines for Medical Education.” (page 6)

The reason why this should be studied has been described in the introduction. In summary, the European Association for Clinical Pharmacology and Therapeutics (EACPT) wants to harmonize the education and training in CPT for all European medical schools, in order to better prepare futures doctors on their task as prescriber. Therefore, we initiated an exam on prescribing knowledge and skills (with the idea of ‘teaching to the test’). However, we do not know which medicines are available and frequently prescribed in Europe, and there is no consensus on which medicines a junior should be able to prescribe safely and effectively. With such a list we can form the exam and education/training in CPT.

Regarding the methodology and the replicability of the study, we are not very sure how to adjust the protocol. Reviewer 2 described: *“I felt the described process could be reasonably replicated”* and reviewer 3 did not mentioned it. Moreover, we described the process of recruitment and creating the drug in detail, and are transparent in our statistics. Therefore, we did not change the protocol. However, if reviewer 1 can describe his concerns in more detail, we are of course likely to adjust the protocol if necessary.

Reviewer 2

First of all, thank you for the positive feedback and comments on our protocol. Also thanks to the thoughtful suggestions. Below you can find our comments.

1. I was perplexed as to why pharmacists were not included (given the multi-professional nature of the prescribing process and their role in surveillance and safety) especially as they have been successfully included in studies where consensus was sort on conditions that junior doctors should be able to prescribe for. I wonder if this had been considered?

We agree with the reviewer that pharmacists should be included as well. While they usually do not prescribe medicines, there are involved in teaching CPT in most medical curricula and play a role in surveillance and safety. Therefore, we included (hospital)pharmacists:

“At least five healthcare professionals, preferably a surgeon (e.g. general surgeon); an internist (e.g. general internist, gastroenterologists, pulmonologist, cardiologist); a general practitioner; a specialist in geriatric medicine or geriatrician; a (hospital)pharmacist” (page 7)

2. I was also unclear about the aimed recruitment targets for the expert response and wondered if this could be clarified? Does 200-250 experts relate to the response rate from coordinators and their expert groups or does it relate to the total number of individual survey respondents (including specialists, junior doctors etc). Obviously this is likely to make a significant impact on the representative nature and consensus achieved. I wonder if this might be clarified?

We agree that the number of expected experts is not described clearly. In previous consensus/survey studies with this group of coordinators, usual 25% (around n= 100) of them complete the study. Assuming that a coordinator recruits 5-6 experts, and assuming a response rate of 25% as well, we expect 100 coordinators and 125-150 additional experts that will complete the whole study. We have included this in the manuscript.

“Based on previous studies with this dedicated group of experts,^{22,26} we expect a response rate of 25% for the coordinators, who will represent all European countries. Assuming 5-6 recruited experts per coordinator and a response rate of 25% as well, we expect in total 200-250 experts to complete the study.” (page 7)

3. The limitations identified seemed reasonable but the papers applicability might to a large part depend on the initial recruitment (ensuring it was representative of the broadest range of European countries) and avoiding participant drop out between surveys (which may be related to elements like survey length etc). I would have welcomed more in the methods as to the implications and how this would be addressed.

We agree that the applicability of the list depends on the number of experts and their diversity (demographics, specialty, teaching experience etc.). Based on our previous studies, we know that all European countries have dedicated experts in this field of research, and therefore we expect that the response rate will be sufficient. We have included this in the manuscript as describe in number 2, see above.

Avoiding participant drop-out between the surveys is a justified concern. The survey is a long list, therefore we will structurally categorize it and we will make it possible that experts don't have to complete the survey at once. We have added this to method section.

“In order to minimize participant drop-out, the list of medicines will be structured and participants are allowed to complete a portion of a survey and return later to finish the remaining part. The list must be completed within two weeks, a reminder will be send after one week.” (page 8)

Reviewer 3

First of all, thank you for the positive feedback and the comments on our protocol. Below you can find our comments on the suggestions.

1. Page 9: The Drug List could be described before the Expert Panel simply to report this in the same order as referred to in Line 9 of page 30 (or change the order here)

We agree with this suggestion. In order to be consistent we will report the section *Drug list* prior to the section *Expert panel*.

2. Page 16 Line 3 - unclear why there is a closing bracket at the end of 'Omeprazole, pantoprazole,esomeprazole)'

We removed the closing bracket.

3. Page 18 Line 22 - unclear why it is called 'Folium acid' when in the previous list it was appropriately called 'Folic acid'

Ee changed it into folic acid.

4. Page 19 Line 26 - I think Zink should be Zinc

We changed it into Zinc

5. Page 19 Line 27 - unclear why a brand (Vaseline) is used instead of the generic (petroleum jelly)

Thank you, we agree that using brand names is inappropriate. We changed it into petroleum jelly.

6. Page 19 - miconazole appears twice under cutaneous (Line 22 and line 39)

We removed one of the lines.